# OCT Angiography in Noninfectious Uveitis: A Description of Five Cases and Clinical Applications

**DOI:** 10.3390/diagnostics13071296

**Published:** 2023-03-30

**Authors:** Samyuktha Melachuri, Kunal K. Dansingani, Joshua Wesalo, Manuel Paez-Escamilla, Meghal Gagrani, Sarah Atta, Chad Indermill, José-Alain Sahel, Ken K. Nischal, Jay Chhablani, Marie-Hélène Errera

**Affiliations:** 1Department of Ophthalmology, University of Pittsburgh, Pittsburgh, PA 15213, USA; 2School of Medicine, University of Pittsburgh, Pittsburgh, PA 15213, USA

**Keywords:** choroidal granuloma, lupus retinopathy, OCT angiography, punctate inner choroiditis, retinal vasculitis, uveitis

## Abstract

Background: Optical coherence tomography angiography (OCTA) is a noninvasive imaging modality used to analyze the retinochoroidal vasculature and detect vascular flow. The resulting images can be segmented to view each vascular plexus individually. While fluorescein angiography is still the gold standard for the diagnosis of posterior uveitis, it has limitations, and can be replaced by OCTA in some cases. Methods: This case series describes five patients with posterior noninfectious uveitis and their description by OCTA. Results: Cases included lupus retinopathy (*n* = 1) for which OCTA showed ischemic maculopathy as areas of flow deficit at the superficial and deep capillary plexus; choroidal granulomas (*n* = 1) with a non-detectable flow signal in the choroid; active punctate inner choroiditis and multifocal choroiditis (*n* = 1) with OCTA that showed active inflammatory chorioretinal lesions as non-detectable flow signals in choriocapillaris and choroid; dense type 2 inflammatory secondary neovascularization (*n* = 1) associated with active choroiditis; and acute posterior multifocal placoid pigment epitheliopathy (APMPPE) (*n* = 1) without flow abnormalities at the superficial and deep retinal plexuses but non-detectable flow at the levels of the choriocapillaris and choroid. Conclusions: Ophthalmologists can use OCTA to identify inflammatory changes in retinal and choroidal vasculature, aiding in the diagnosis, management, and monitoring of posterior uveitis.

## 1. Introduction

Optical coherence tomography angiography (OCTA) is a noninvasive imaging modality used to analyze retinal and choroidal vasculature via phase decorrelation within vessels in sequential images and represent the changes between images as vascular flow [1]. It produces maps of where the OCT pixel intensities change over time. These maps, to a great extent (adjusting for various artifacts), correspond with the morphology of the vasculature. More precisely, this technology depends on a multitude of OCT scans in the same region at speeds of up to 100 kHz and assumes that no structural changes occur between scans [2]. Therefore, any change between scans is interpreted by OCTA as “movement” or blood flow. OCTA images are usually examined in an en face orientation after segmenting the volumes with respect to a chosen tissue layer. The reference layer is chosen based on the lesion types being studied.

Spectral-domain (SD) OCTA and swept-source (SS) OCTA have been found useful to study inflammatory chorioretinal diseases [3,4,5,6,7,8]. For example, inflammatory chorio retinal neovascularization (iCNV) can be imaged with OCTA, as it yields high-resolution images of abnormal vasculature [9]. OCTA is both fast to obtain and does not use any injectable dye. While OCTA cannot detect “leakage,” the hallmark of inflammation due to changes in the blood–retinal barrier, OCTA can differentiate iCNV from other inflammatory lesions (e.g., such as those of punctate inner choroiditis or multifocal choroiditis) by visualizing the neovascular networks in iCNVs. OCTA also can identify areas of retinal nonperfusion and reperfusion over time. Other uses of OCTA in uveitis are more anecdotal, such as its use to detect dilation of vessels during acute iridocyclitis, the increase in vessel density in the optic disc during acute inflammation, or the decrease in capillary density and complexity in both superficial and deep vascular plexuses during chronic inflammation [10]. There are currently no widely accepted clinical guidelines regarding the use of multimodal imaging for retinal and chorioretinal disorders that can arise during uveitis, especially because the diagnosis and measures of uveitis activity are disease-specific, making it challenging to implement universal guidelines.

With the advent of new technologies, the need to standardize interpretation into a ubiquitous, easy-to-use language becomes paramount. For this purpose, the consensus in the nomenclature for reporting optical coherence tomography angiography (OCTA) findings in uveitis established guidelines for the definition of wide-field OCTA (WD-OCTA) as greater than 70 degrees of view. It also established the terms “flow deficit” and “non-detectable flow” that the expert panel explains describing “abnormal OCTA signal due to slow flow and vessel displacement, respectively” even given that OCTA does not directly measure the speed of flow, and the same panel recommends the term “loose” and “dense” to describe the appearance of inflammatory choroidal neovascularization. These unified terms are used throughout this paper to describe the OCTA findings in our patient cohort [11].

We aim to illustrate the utility of OCTA in monitoring inflammatory eye diseases through our experience in five patients with posterior noninfectious uveitis and demonstrate the contribution of OCTA findings to clinical decision making in these cases.

## 2. Materials and Methods

We selected five patients who were diagnosed with posterior uveitis and evaluated with OCTA (Zeiss PLEX^®^ Elite 9000 using a swept-source tunable laser, with a center wavelength between 1040 and 1060 nm as the optical source, or Heidelberg Spectralis^®^ OCT2 using 840 nm) at the initial presentation. The Zeiss Plex Elite operates at a speed of 100,000 A-scans/sec; our device has not yet had the 200,000 Hz upgrade. The Heidelberg Spectralis^®^ OCT2 operates with a A-scan rate of 85,000 Hz.

This study was conducted in accordance with the tenets of the Declaration of Helsinki and approved by University of Pittsburgh Medical Center’s (UPMC) Institutional Review Board (Number: STUDY19030187—Multimodal Imaging in Ophthalmology). They had presentation and follow-up at the UPMC Eye Center from January 2020 to September 2022. Patients were selected for this study based on established uveitis literature that inflammatory disease presents with a variety of characteristic lesion types, including vascular permeability, vasculitic ischemia, inflammatory lesions and nodules, infiltrates, neovascularization, and other tissue reactions [3]. The cases were also selected to illustrate lesions of the superficial capillary plexus, choriocapillaris and choroid. The patients demonstrate a variety of OCTA findings related to these tissue-level phenomena. When automated segmentation failed, manual segmentation was used to overcome this limitation and to maximize the yield of usable information from OCTA scans.

An important concept explained in our paper, which is essential to the correct interpretation of reduced flow signal on OCTA, is that distinguishing between masking, and genuinely reduced flow requires side-by-side comparison with the equivalent slice from the structural OCT.

To interpret the OCTA pictures, we compared them with corresponding en face structural OCT pictures. When OCTA showed no flow, we used the corresponding en face structural OCT image to evaluate whether a shadow from an overlying pathology could be detected. Shadowing is caused by an obstacle to light penetration through the retina which is obscured by an overlying pathology. This reduction in signal is termed shadowing and can be misinterpreted as reduced flow in OCTA images.

We used terminology from ‘Consensus-based recommendations for optical coherence tomography angiography reporting in uveitis’ to describe OCTA findings. Specifically, we applied Pichi et al.’s nomenclature in cases of OCTA signal attenuation, we used the term “flow deficit”. In cases of OCTA vascular flow decrease due to blockage of the signal by inflammatory cells and granulomas in the choriocapillaris/choroid confirmed by indocyanine green angiography, we used the term “non-detectable flow signal”. In the OCTA description of inflammatory secondary neovascularization, we used the terms “loose” or “dense” [11].

We relied on automated segmentation proprietary to each device (closed source). These are based on pre-set boundary curves, such as the vitreo–retinal interface, outer retina, RPE, choriocapillaris. If there were segmentation errors in a particular curve set, we chose the next closest curve set which did not have segmentation errors and which approximated the contours of the tissue feature of interest. We were then able to adjust the global z-position of this curve set to window in on the tissue feature of interest.

The choroidal thickness (CT) measurements were obtained from manual segmentation of OCT B-scans at the fovea for case numbers 1, 2, 4 and 5 and at the location of the choroiditis lesion (for case 3). Follow-up CT measurements were available for case numbers 1, 3 and 4. The Standardization of Uveitis Nomenclature (SUN) criteria for diagnosis of uveitis is a standardized grading system of the anterior chamber, structural complications of uveitis, outcomes of uveitis, and visual acuity [12,13]. Uveitis workup included chest X-ray to detect pulmonary sarcoidosis or tuberculosis, syphilis serologies, and Lyme serology, if suspected. Because the uveitis met criteria of severity due to posterior involvement and chronicity, the patient was referred to Rheumatology to complete the uveitis workup if there was clinical or laboratory suspicion of autoimmune or infectious etiology, respectively. Etiology of uveitis can include infection, systemic immune-mediated disease, drugs and hypersensitivity reactions, or genetic predisposition [12].

## 3. Results

The demographic, clinical data, diagnosis, OCT and OCTA findings, and treatment decisions for each patient aided by OCTA are outlined in Table 1.

### 3.1. Case Number 1

A 52-year-old female was seen at a follow-up visit for monitoring the extent of retinal ischemia and vasculitis in her right eye, secondary to lupus retinopathy. Her systemic lupus (with lupus nephritis) was managed with mycophenolate mofetil 2500 mg daily, hydroxychloroquine 4.7 mg/kg daily since the last 5 years, and topical steroids. Fundus examination showed telangiectatic vessels suspicious for early preretinal neovascularization in the macula at the border of the area of retinal ischemia (Figure 1A), which was identified previously by fluorescein angiography (FA) (Figure 1B). The patient had previously presented for multiple episodes of intravitreal hemorrhages from telangiectatic vessels. Each vitreous hemorrhage was resolved with intravitreal anti-VEGF. Sub-Tenon periocular steroid (triamcinolone) injections were performed twice at 3-month intervals to quiet the eventual underlying inflammation that might be responsible for telangiectatic vessels bleeding. Plex Elite OCTA (12 mm *×* 12 mm) demonstrated an area of “flow deficit” or decreased signal at the level of superficial (SCP) (Figure 1C) and deep capillary plexus (DCP) (Appendix A) corresponding to retinal ischemia in the posterior pole. The area of flow deficit observed on OCTA was of similar size to the retinal ischemia detected on FA 2.5 months prior and at presentation 2 years previously. The border of the nonperfused area was seen on the en face structure. The telangiectatic vessels were not detected on the en face structure (Figure 1D,E). A multifocal electrophysiology showed that the responses in the fellow eye, left eye were within normal limits, indicating normal macula function. There is no generalized or focal decrease in response amplitude or implicit time suggestive of hydrochloroquine toxicity. In the right eye, the responses were reduced temporally consistent with the lupus retinopathy and consequent visual field deficit observed.

OCTA acquisition was continued at each visit every 3 to 4 months during 29 months to monitor the area of ischemic maculopathy and to guide treatment. No change in flow deficit was detected (Figure 1C,F), so the immunosuppression therapy was continued. OCTA did not allow for description of the size of capillary telangiectatic vessels.

Appendix A is an example for CT measurements. The choroidal thickness measurements were obtained from manual segmentation of OCT B-scans at the fovea at baseline, +11 months, +21 months and +32 months.

### 3.2. Case Number 2

A 74-year-old woman with a history of pulmonary sarcoidosis on oral hydroxychloroquine daily dose of 1.9 mg/mg/day started 18 months ago was referred for posterior uveitis. She presented with multiple deep choroidal lesions in the posterior pole of the right eye (Figure 2A), and the corresponding lesions were seen as hypofluorescent lesions on indocyanine angiography (ICG) corresponding to choroidal granulomas (Figure 2B). Because her vision was 20/20 in the right eye, without visual symptoms, we decided to observe. OCTA was performed at 6-week follow-up to monitor the lesions, and it showed areas of “non-detectable flow signal” in the corresponding inner choroid (Figure 2C); the size and location of the choroidal granulomas on OCTA images were unchanged in both eyes as compared to baseline ICG. SD-OCT was performed that showed unremarkable outer retinal layers, ruling out hydroxychloroquine toxicity; the 10-2 Humphrey visual fields were unreliable.

### 3.3. Case Number 3

A 41-year-old female with a history of presumed ocular histoplasmosis syndrome (POHS) in the right eye with a history of intraocular immunomodulatory treatment and anti-VEGF injections presented with blurry vision in the left eye. The dilated fundus exam was significant for bilateral vitreous clumps, bilateral peripapillary atrophy, and several small punched out choroidal lesions in the macula of the right eye. Fundus autofluorescence (FAF) showed multiple hypoautofluorescent (hypoAF) spots in the posterior pole corresponding to old choroiditis scars and one active hyperAF lesion superior to fovea in the left eye (Figure 3A). An OCT scan showed a small hyper-reflective lesion above the RPE along with discontinuity of outer retinal structures with mild focal thickening of retina (Figure 3B). To distinguish between an active punctate inner choroiditis (PIC) lesion and an inflammatory choroidal neovascular membranes (CNVM), a Plex Elite OCTA (6 mm *×* 6 mm) was performed and demonstrated a dense neovascular network capillary branching resembling a fine-net pattern occurring at the level of choriocapillaris (Figure 3C), which was also hyperreflective on en face structural OCT (Figure 3D). OCTA findings allowed us to conclude that the offending lesion was not just inflammatory but that it also contained a neovascular network. For this reason, the patient was treated not only for inflammation with oral prednisone starting at a dose of 40 mg daily and oral mycophenolate mofetil 250 mg twice a day but also with anti-VEGF injections of bevacizumab for neovascularization. At the last follow-up at 3 months, the neovascular lesion had regressed on OCT and SD-OCT (Figure 3E,F). The lesion is smaller on the structure OCT (Figure 3G) and OCTA at 3-month follow-up (Figure 3F) showing less flow signal and less reflectivity after treatment as compared to before on Figure 3C,D. OCTA imaging was repeated 2-monthly for 8 months afterwards, and the images remained unchanged.

### 3.4. Case Number 4

A 78-year-old female with a past ocular history of retinal detachment in the right eye was referred by her ophthalmologist for a few weeks of blurry vision in the left eye and photophobia and was found to have acute panuveitis in the left eye. Visual acuity was 20/200 in the left eye. The slit-lamp exam was significant for 2+ cell in the anterior chamber and 3+ cell in vitreous of the left eye. The right eye vision was reduced to light perception with silicone oil fill and attached retina on fundus exam. The dilated fundus exam showed multiple punched-out chorioretinal lesions in the retinal periphery of the left eye. OCTA showed multiple lesions that presented as “non-detectable flow signal” at the posterior pole at the level of choriocapillaris in the left eye (Figure 4A). SD-OCT of the left eye showed multi-inflammatory lesions in the RPE and choriocapillaris (Figure 4C). She was started on oral prednisone 0.75 mg/kg/day and topical prednisolone (left eye) with a subsequent steroid taper for active multifocal choroiditis. Laboratory results were significant for PR-3 antibodies, which was concerning for ANCA-associated vasculitis syndrome. Consequently, she was initiated on oral methotrexate weekly. OCT one month after the initial presentation showed worsening of the lesions in the RPE and choriocapillaris. Subsequently, the methotrexate dose was increased to 15 mg weekly. At her 3-month follow-up appointment, new active choroiditis lesions were seen (Figure 4D). The prednisone dose was increased, and adalimumab (subcutaneous) was initiated. However, as the steroids were tapered, OCTA showed worsening findings, more elevated chorioretinal lesions with vascular flow, corresponding to inflammatory CNVM (Figure 4G). The patient received an intravitreal dexamethasone injection in the left eye, and the frequency of adalimumab injections was increased to weekly. Images are shown in Figure 4.

### 3.5. Case Number 5

A 17-year-old female was referred by pediatric ophthalmology for blurry vision and loss of central vision in her left eye for 1 month. On exam, her visual acuity was 20/20 in her right eye and 20/80 in her left eye without improvement with pinhole or refraction. She had a monocular nasal field cut. On fundus exam, the left eye had deep placoid subretinal lesions between the optic nerve and fovea and retinal pigment epithelial pigment changes at the fovea (Figure 5A). The right eye had deep small white dots outside the vascular arcades. The exam was not significant for anterior or posterior chamber inflammation. Fundus autofluorescence showed a hypoautofluorescent lesion with a hyperautofluorescent margin at the center of the left eye and one lesion supero-nasally in the left eye (Figure 5B). SD-OCT showed outer retinal reflectivity with ellipsoid zone disruption at the macula of the left eye. The clinical picture and imaging were typical for acute posterior multifocal placoid pigment epitheliopathy (APMPPE) in the left eye. The patient was started on 0.75 m/kg/day of prednisone daily due to macular involvement. After 3 weeks of a slow steroid taper, vision in the left eye improved to 20/30. Fundus photos showed the resolution of white lesions in the right and left eyes and an unresolved pigmented lesion deep in the macula of the left eye from 3 weeks prior. SD-OCT showed resolved hyperreflectivity in the ONL layer but continued alteration of the ellipsoid zone (Figure 5C). OCTA was significant for non-detectable flow signal at the macula (Figure 5D) in the location of the pigmented macular lesion in the fundus photo.

## 4. Discussion

In this case series, we described the utility of OCTA in the diagnosis, management, and monitoring inflammatory biomarkers of five patients with posterior noninfectious uveitis. In our experience, OCTA is an excellent tool for (1) retinal and (2) choroidal nonperfusion, (3) choroid occupying lesions and (4) CNV, which are all inflammatory biomarkers.

(1) OCTA images can detect and assess the extent of retinal nonperfusion in occlusive retinal vasculitis. It visualizes the disruption or absence of normal capillary plexuses (SCP, DCP), which is described as a “flow deficit” in affected retinal areas in various uveitic entities [11]. These entities include infectious retinitis (acute retinal necrosis secondary to Herpes Simplex Virus and Varicella Zoster Virus; Cytomegalovirus retinitis), lupus retinopathy, idiopathic retinal vasculitis, retinal vein occlusion in Behçet disease, retinal vein or artery occlusion in ocular sarcoidosis, and tuberculosis, among others. Case 1, whose right eye was affected by lupus retinopathy, presented with a typical ischemic retinal area with a decreased flow rate in the SCP and DCP. OCTA better characterized and monitored the extent of the ischemic maculopathy over time compared to en face structure OCT. Interestingly, OCTA provided sufficient information for FA not to be needed at this time point.

(2) OCTA allows for the detection of “flow deficit” due to vascular occlusion at the level of the choriocapillaris and at the level of choroid. Choriocapillaris hypoperfusion presents with disruption and atrophy of the RPE and outer retinal layers, which are changes that can be captured by OCTA [14]. In our case series, case number 5 diagnosed with APMPEE presented as expected for that condition with choriocapillaris hypoperfusion as confirmed with OCTA.

(3) OCTA appears also useful to detect choroid occupying lesions. Choroidal granulomas appear as areas of “non-detectable flow signal” on OCTA as they compress the surrounding vasculature and impair choroidal blood flow. We used OCTA to monitor the number and size of choroidal granulomas due to sarcoidosis in case 2. These inflammatory lesions in the choriocapillaris and inner choroid were easily detectable on OCTA compared to the visualization via ICG.

Patients with MFC typically have round-shaped hyperreflective lesions that present with focal elevation of the RPE, focal atrophy of the outer retina and/or RPE, or the presence of sub-RPE hyper-reflective deposits on SD-OCT. They can also display focal hyper-reflective dots in the inner choroid [15]. OCTA can differentiate these inflammatory foci from iCNVs that complicate MFC and PIC in 32–46% of patients [16,17,18]. In fact, MFC lesions do not show any internal flow on OCTA. In contrast, blood flow is clearly visible within the neovascular network in patients with iCNV [19]. The active MFC lesions in Case 4 presented as non-detectable flow signal that increased in size when the disease was inadequately controlled. Then, OCTA detected choroidal neovascularization as a dense vascular flow increase. The small amount of SRF seen on SD-OCT confirmed the CNV lesion. In case number 3, OCTA detected inflammatory neovascularization as a dense neovascular network resembling a fine-net pattern occurring at the level of choriocapillaris. Before and after anti-VEGF treatment, on the en face structure and OCTA, the CNV appears smaller, suggesting a flow signal before and lack of flow signal after treatment. In Case 3, the corresponding SD-OCT was unable to differentiate active choroiditis from CNV.

It is of note that the en face structure gave the same information as OCTA in the detection and monitoring of inflammatory choroidal neovascular membranes under appropriate treatment and in choriocapillaris flow reduction or ischemia defects in APMPPE lesions.

We found that the Pichi et al. nomenclature was effective for describing the findings: the terms ’flow deficit’ to describe abnormal OCTA flow signal secondary to slow flow to measure retinal ischemia in the case of occlusive retinal vasculitis (case number 1) and ’non-detectable flow signal’ to describe vessels displacement secondary to choroidal granulomas, multifocal choroiditis or placoid lesion in APMPPE (case numbers 2, 4 and 5). We used the term ‘dense’ to describe the appearance of inflammatory choroidal neovascularization (case number 3) [4].

We would like to discuss the advantages and disadvantages of OCTA versus current imaging modalities such as FA and ICG for the diagnosis and management of uveitis. In our experience, the value of OCTA vs. FA is mainly in distinguishing between vascular lesions and inflammatory lesions.

OCTA provides useful information such as detecting a “flow void” due to slower than normal flow in the SCP in autoimmune retinopathy and idiopathic retinal vasculitis [20]. FA is a dynamic study that gives useful information in a two-dimensional fashion showing details primarily comprised of the superficial retinal capillary plexus. However, FA image interpretation depends on dynamic properties such as dye leakage, staining, window defect, and blockage [21], whereas OCTA is non-invasive and provides static volumetric angiographic information depicting a snapshot in time of blood flow. OCTA provides highly detailed images of flow in the superficial, intermediate, and deep retinal capillary plexuses as well as the radial peripapillary network and choriocapillaris [22,23].

ICG angiography is the gold standard to detect choroidal granulomas; however, it carries the risk of allergy, and therefore, OCTA is favored in routine practice specially to monitor the appearance and size of granulomas. When the OCTA signal is altered by space-occupying lesions, such as sarcoid granulomas in the choroid, the flow signal is undetectable. Around the BM-RPE complex, OCTA can detect the presence of neovascularization within what otherwise might appear as inflammatory lesions, but we do not know the sensitivity or specificity for this. The benefit of OCTA in detecting inflammatory choroidal neovascular membranes is avoiding the use of FA or ICG. OCTA can be safer than serial retinal angiographies to follow treatment response in eyes with inflammatory neovascular membranes, especially in patients affected by PIC and MFC [24,25]. A review by Ahnood et al. found the rate of inflammatory neovascular membrane in patients with PIC to be 22% [26]. Invernizzi et al. have shown the superiority of OCT to detect early variations in the size of choroidal granulomas in response to treatment [27]. However, in our experience, choroidal granulomas in sarcoidosis are difficult to visualize with SD-OCT and are more clearly visualized with OCTA as “non-detectable flow signal” areas [14]. The pathophysiology of APMPPE and MEWDS is different. APMPPE is a “primarily choroidal inflammatory disorder leading to outer retinal involvement” because en face OCT scans at the level of the choriocapillaris in APMPPE show choroidal infiltration and dilation of choroidal vessels [28]. Some authors have shown evidence of inner choroidal or choriocapillaris flow reduction or ischemia defects in APMPPE lesions [29,30]. MEWDS, on the other hand, is an epitheliopathy with RPE involvement but without choriocapillaris involvement on OCTA [28].

In terms of limitations of OCTA, OCTA does not detect vascular leakage, which details the level of inflammation of retinal vasculitis and which is also one of the more common features we look for as inflammatory vasculopathy [31]. Uveitis specialists still rely on FA, as it is extremely sensitive for detecting retinal vessel inflammation via vascular leakage, but it requires dye, which has a potential for allergic reactions [31]. Most OCTA machines provide a limited view, 12 mm × 12 mm of the posterior pole, and they cannot be used to study the peripheral fundus, although “WF-OCTA” can be obtained by montaging scans. The OCTA signal is altered by different artifacts such as media opacities and by fluid in pathologic conditions such as cystoid macular edema in the SCP, leading to a “non-detectable flow signal” in that location. Vascular flow can only be detected within a certain range of velocities, preventing the accurate visualization of vascular networks outside this range [14]. There is limited use of OCTA in anterior uveitis (AU). Nevertheless, Pichi et al. used OCTA to detect a dilation of the iris vessels in acute AU and resolution with treatment. As the vitreous has no vasculature, the use of OCTA in this area is restricted. However, this technique can visualize neovascularization into the vitreous [14,31].

Moreover, significant inter-instrument differences in density measurements of the superficial capillary plexus have been demonstrated by previous studies. They proposed that variations in manufacturer-stated segmentation boundaries may be one reason for these differences.

Several published studies have compared the two OCTA devices that we used, the Heidelberg Spectralis OCT2 module and Zeiss Plex Elite 9000 Swept-Source OCT. Lu et al. have compared a smaller section 3 mm × 3 mm, and they showed that the foveal avascular zone area showed no significant difference across devices with an ICC of >95%. However, there were statistically significant differences for SCP and DCP vessel densities. They found also a statistically significant difference for fractal dimension at the DCP layer [32]. Another study has found a sensitivity of 96.3% for the PlexElite device and 87.9% for the Heidelberg OCT2 for the detection of macular neovascularization (MN) in neovascular age macular degeneration (ref). However, in uveitis, most of the MNVs are of type 2, and the same study found that 100% of this type of MNVs was detected by all OCTA devices independently of the SD or SS technology. They found that undetectable/undetected MNV can represent up to 45% of the examinations [33].

Despite the limitations, we showed that OCTA plays an increasingly important role in the diagnosis and monitoring of treatment response in noninfectious posterior uveitis and can be used in combination with FA, ICG, and OCT. We recommend a multimodal approach to visualize the choroidal and retinal vasculature in inflammatory processes [14]. Of note, previous studies have shown that OCTA has the unique ability to demonstrate pathological flow impairment at the level of choriocapillaris in infectious posterior uveitis such as in active tubercular SLC. Particularly, Agarwal et al. showed that during longitudinal follow-up, in the multifocal type of SLC, the mean lesion area decreased in all eyes compared with baseline [34].

## 5. Conclusions

OCTA is an effective, noninvasive imaging modality that provides greater clarity and quantity of information than many of the existing imaging techniques. Ophthalmologists can use OCTA to identify inflammatory biomarkers (CNV, retinal nonperfusion, choroidal nonperfusion and choroid space occupying lesions such as granulomas) aiding in the management and monitoring of uveitis. Therefore, the value of OCTA vs. FA is mainly in distinguishing between vascular lesions and inflammatory lesions, aiding in the diagnosis.

## Figures and Tables

**Figure 1 diagnostics-13-01296-f001:**
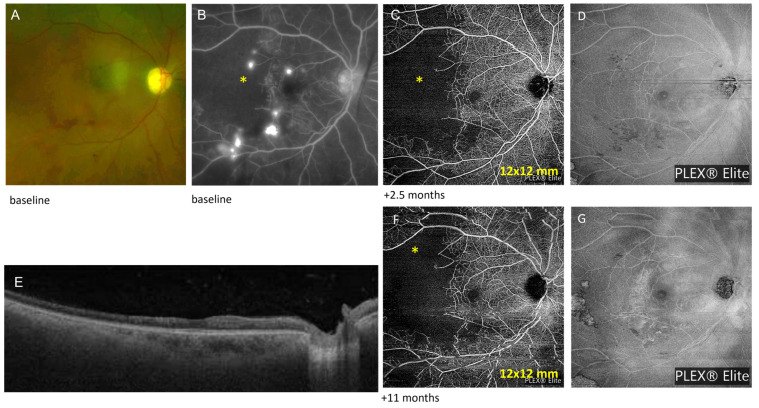
Case 1: Ischemic retina in lupus retinopathy. (**A**) Fundus photo showing ischemic maculopathy with telangiectatic vessels (abnormal prominent capillaries) at its border in the right eye; (**B**) Fluorescein angiography (intermediate phase) showing a macular area of capillary non-perfusion (asterisk) and hyperfluorescent telangiectatic vessels at its border; (**C**) Zeiss PLEX^®^ Elite 9000 OCTA (12 mm × 12 mm) demonstrating an area of “flow deficit” or decreased signal in the superficial capillary plexus (SCP) corresponding to retinal ischemia in the temporal area (asterisk); (**D**) The border of nonperfusion is seen on the structural OCT; (**E**) SD-OCT showing foveal thinning; (**F**) Eleven-month follow-up OCTA that shows no change in flow deficit; (**G**) The border of nonperfusion is seen on the corresponding structural OCT.

**Figure 2 diagnostics-13-01296-f002:**
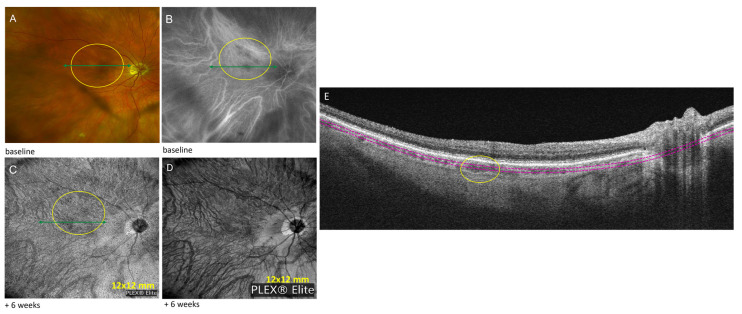
Case 2: Sarcoidosis: Choroidal granuloma. (**A**) Color fundus photograph shows multiple sarcoid choroidal granulomas that appear as deep choroidal lesions in the posterior pole (circle); (**B**) ICG angiogram shows corresponding lesions as hypofluorescent lesions (circle); (**C**) Zeiss PLEX^®^ Elite 9000 OCTA (12 mm × 12 mm) performed 6 weeks later showed the granulomas as areas of “non-detectable flow signal” in the corresponding inner choroid (circle). (**D**) On the en face structure, we do not see a shadow in the lesion. Green lines in (**A**–**C**) correspond to the position of the SD-OCT scan in E. (**E**) Choroidal granuloma occupying the full-thickness of the choroid on SD-OCT (circle). En face segmentation boundaries shown in magenta.

**Figure 3 diagnostics-13-01296-f003:**
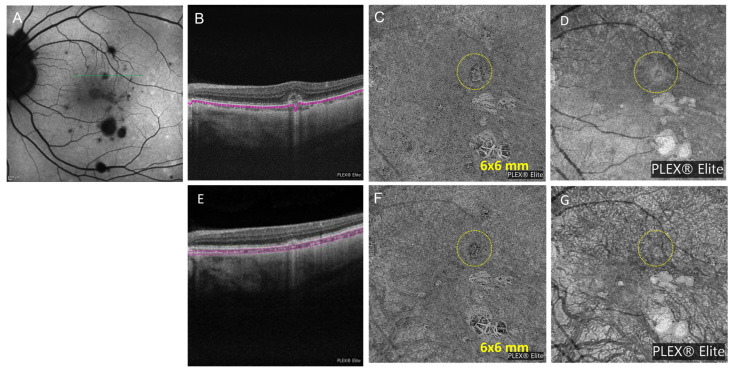
Case 3: Punctate inner choroiditis and inflammatory neovascular membrane. (**A**) On initial presentation, fundus autofluorescence (FAF) showed multiple hypoautofluorescent spots in the posterior pole corresponding to old choroiditis scars and one active hyperautofluorescent lesion superior to fovea in the left eye; (**B**) OCT showed retinal pigment epithelium (RPE) elevation with loss of ellipsoid zone without intra retinal fluid (IRF) in the left eye (en face segmentation boundaries in magenta); (**C**) A Plex Elite OCTA (6 mm × 6 mm) demonstrated a dense neovascular network capillary branching resembling a fine-net pattern occurring at the level of choriocapillaris (circle); (**D**) The corresponding en face structure showed the same hyperreflective lesion (circle); (**E**) Three months after initiation of steroid therapy, OCT showed resolution of retinal pigment epithelium (RPE) elevation without intraretinal fluid (IRF) in the left eye; (**F**) A Plex Elite OCTA (6 mm × 6 mm) demonstrating a regressed neovascular network at the level of choriocapillaris after intravitreal bevacizumab (circle); (**G**) The lesion shows less reflectivity than before on the corresponding en face structure (circle).

**Figure 4 diagnostics-13-01296-f004:**
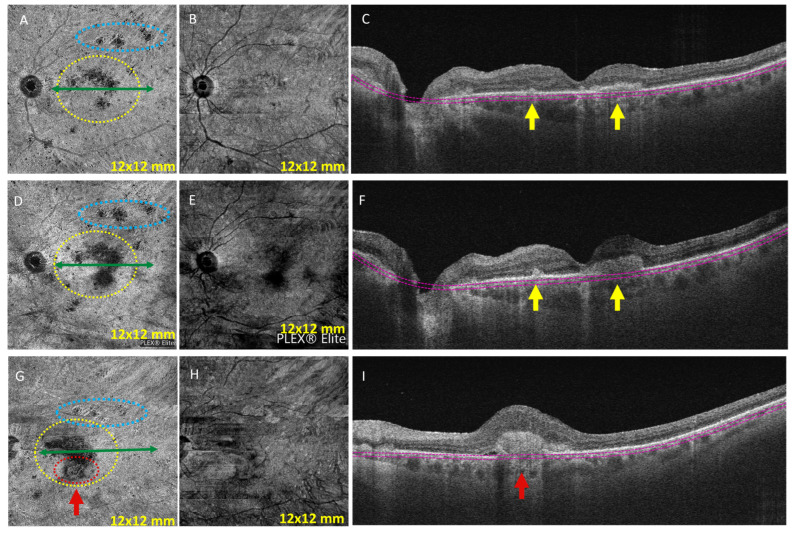
Case number 4: Multifocal choroiditis and secondary inflammatory neovascularization. (**A**) Plex Elite OCTA (12 mm × 12 mm) showed multiple lesions that presented as “non-detectable flow signal” at the posterior pole at the level of choriocapillaris in the left eye (lesions into yellow and blue circles). (**B**) The structure en face confirms that this is not shadow, that it is choriocapillaris non-perfusion with preserved reflectivity. (**C**) Corresponding SD-OCT showed multi-inflammatory RPE and choriocapillaris lesions in the left eye (yellow arrows). (**D**) Zeiss PLEX^®^ Elite 9000 OCTA (12 mm × 12 mm) performed 3 months after initial adalimumab injection (subcutaneous every 2 weeks) and after tapering a course of oral steroids showed larger areas of non-detectable flow signal (yellow and blue circles). (**E**) Structure en face uninterpretable. (**F**) Corresponding SD-OCT showed multiple more highly elevated inflammatory RPE and choriocapillaris lesions in the left eye (yellow arrows). (**G**) Four months after the initial adalimumab injection (s/p 3 adalimumab injections weekly), the non-detectable flow signal area in the yellow circle is still visible but with a new dense inflammatory secondary neovascularization at the bottom of the lesion (red circle). (**H**) Structure en face uninterpretable. (**I**) Corresponding SD-OCT showed higher inflammatory secondary neovascularization as subretinal hyperreflective material (red arrow) with a small amount of subretinal fluid in the left eye. Manual segmentation used for boundaries in OCT (pink lines in images (**C**,**F**,**I**)).

**Figure 5 diagnostics-13-01296-f005:**
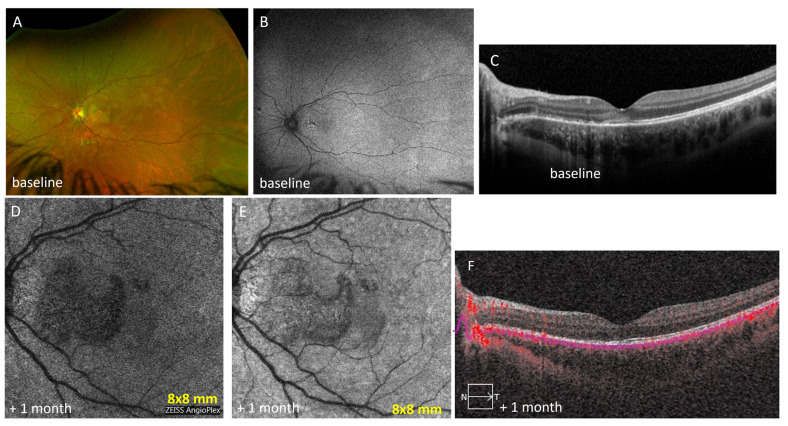
Case 5: APMPPE. (**A**) Wide-field fundus photo of the left eye showing deep placoid subretinal lesions between optic nerve and fovea and retinal pigment epithelial pigment changes at the fovea. (**B**) Wide-field fundus autofluorescence picture of left eye showing a hypoautofluorescent lesion with a hyperfluorescent margin at the center of the left eye and one lesion supero-nasally in the left eye. (**C**) SD-OCT (Heidelberg Spectralis) showed outer retinal reflectivity with ellipsoid zone disruption at the macula of the left eye. There are also corresponding hyperreflective lesions in the choriocapillaris. (**D**) Zeiss PLEX^®^ Elite 9000, OCTA (9 mm × 9 mm) at the level of choriocapillaris after one month was significant for non-detectable flow signal at the macula. (**E**) The corresponding structure shows that there is a shadow corresponding to the macular lesion. (**F**) SD-OCT shows an outer retina alteration nasal from the fovea at the lesion’s location. Due to frame averaging, the OCT image provided by Spectralis Heidelberg SD-OCT (image (**C**)) had a better signal to noise ratio than the OCT provided by the SS-OCT Zeiss PLEX^®^ Elite 9000 at follow-up (image (**F**)).

**Table 1 diagnostics-13-01296-t001:** Description of diagnosis, imaging findings, and management of the five cases included in the case series.

Cases	Diagnosis	OCT		OCTA	Choroidal Thickness (µm)	
1	Lupus retinopathy	Foveal thinning in the right eye, normal anatomy in the left eye		Zeiss PLEX^®^ Elite 9000 OCTA (12 mm × 12 mm): area of flow deficit or decreased signal in the SCP corresponding to retinal ischemia in the temporal area	baseline: 335 +11 months: 283 +21 months: 271 +32 months: 304	
2	Sarcoid choroidal granulomas	Hyper-reflective choroidal material: choroidal granulomas without subretinal or intraretinal fluid		Zeiss PLEX^®^ Elite 9000 OCTA (12 mm × 12 mm): non-detectable flow signal in corresponding choroid; One month post-injection: size decrease Eleven months post-injection: no changes	Baseline: 220	
3	Punctate inner choroiditis and inflammatory choroidal neovascular membranes	RPE elevation with loss of ellipsoid zone without IRF in the left eye		Heidelberg Spectralis OCTA (6 mm × 6 mm): dense neovascular network capillary branching resembling a fine-net pattern occurring at the level of choriocapillaris	baseline: 144 +1 month: 151 +3 months: 152 +7 months: 137	
4	Multifocal choroiditis	Multi-inflammatory RPE and choriocapillaris lesions in left eye		Zeiss PLEX^®^ Elite 9000 OCTA (12 mm × 12 mm): non-detectable flow signal at posterior pole in choriocapillaris and choroid in left eye	baseline: 398 +3 months: 401 +4 month: 390 +6 months: 415 +7 months: 339	
5	Acute posterior multifocal placoid pigment epitheliopathy	Outer retinal reflectivity with ellipsoid zone disruption at the macula with corresponding hyperreflective lesions in the choriocapillaris in left eye		Zeiss PLEX^®^ Elite 9000 OCTA 9 mm × 9 mm): non-detectable flow signals at the macula	baseline: 371	
**Cases**	**Age/Sex**	**Diagnosis**	**Uveitis Laterality**	**Treatment Decision**	**BCVA**	**Current Tx**
1	52 F	Lupus retinopathy	mono	Intravitreal anti-VEGF Subtenon triamcinolone injections	20/25	Prednisolone acetate drop to right eye daily
2	74 F	Sarcoid choroidal granulomas	mono	Subtenon triamcinolone injection; adalimumab 40 mg subQ every two weeks	20/20	Hydroxyquinoline 200 mg daily
				Intraocular immunomodulatory therapy discontinued		
3	41 F	PIC and inflammatory CNVM	mono	Methylprednisolone 500 mg IV for 3 days; prednisone starting dose: 40 mg daily after IV steroid course; Mycophenolate mofetil 250 mg twice a day; Bevacizumab injections in left eye	20/25	Mycophenolate mofetil 500 mg twice a day
4	78 F	Multifocal choroiditis	mono	Increased prednisone dose to 15 mg daily; brimonidine eye drops in left eye three times a day; adalimumab 40 mg subQ every two weeks; intravitreal dexamethasone injection in left eye	20/60	Methotrexate 15 mg weekly, oral prednisone 10 mg daily, adalimumab 40 mg weekly
5	17 F	APMPPE	mono	Prednisone 0.75 mg/kg/day with 3 weeks of slow steroid taper	20/30	No current treatment

The mean CT was not significantly reduced at last follow-up compared with baseline (mean 292 µm ± 132 at baseline vs. mean 260 µm ± 108) (*p* = 0.75). Appendix A includes all different layers: superficial, deep capillary plexuses, choriocapillaris, choroid to compare with each analyzed patient. Abbreviations: VEGF = vascular endothelial growth factor, SCP = superficial capillary plexus; RPE: retinal pigment epithelium; subQ = subcutaneous.

## Data Availability

Clinical data and images were presented. The original data are subject to patient privacy and regulated by HIPAA. For clinical collaboration, or further case discussion, data can be shared through approved secure mechanisms, after contacting the authors directly.

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
