# Peer review of "OCT Angiography in Noninfectious Uveitis: A Description of Five Cases and Clinical Applications"

_diagnostics, 2023, doi:10.3390/diagnostics13071296_

Round 1

Reviewer 1 Report

The authors describe five cases of non-infectious posterior uveitis studied by OCT-angiography (OCTA). They conclude that this method is useful for the diagnosis of ocular vascular lesions in this disease.

Comments:

1)      A title such as "OCT-angiography in noninfectious uveitis: description of five cases and clinical applications" in my opinion would be clearer and more informative

2)      The authors should specify in the background section of the abstract the unmet needs related to the study of uveitis that may justify adding the use of OCTA. They should also specify in this section that this is a series of 5 clinical cases with non-infectious posterior uveitis.

3)      In the study, patients were studied with three different instruments for OCTA. It would be helpful to know if there are differences in the sensitivity and specificity of the results obtained with different equipment.

4)      Patient 3 is described as having histoplasmosis uveitis. Since the authors consider patients with noninfectious uveitis, this point should be clarified

5)      Although the clinical cases are described in detail throughout the text, Table 1 should be completed with demographic data of the patients (age, sex etc.), bilaterality or monolaterality of uveitis, BCVA, current therapy, to make the reading of the table more informative.

6)      Two patients were on hydroxychloroquine therapy. Given the retinal toxicity of this drug, a comment on this is useful

7)      Authors should provide some data on patient monitoring (frequency of visits, any changes over time possibly detected by OCTA

8)      The advantage of OCTA over other methods that explore the retinal vasculature (fluorescein angiography, indocyanine angiography, enhanced-depth imaging-OCT) should be better commented. The authors should also comment on whether OCTA is sufficient alone or should necessarily be combined with other diagnostic methodologies in the cases studied.

9)      It is also useful to comment on whether this method might be useful also in infectious forms of uveitis

10) Approval of the study by the ethics committee should also be specified in the materials and methods section

Author Response

OCTA Publisher Comments

Reviewer Panel

Diagnostics journal

March 5th, 2023

Dear Reviewers,

Thank you for your helpful comments and suggestions for our manuscript, Manuscript ID: diagnostics-2204641. Please find attached a revised version of the manuscript with Track Changes enabled. Your comments are also addressed as follows:

REVIEWER 1 COMMENTS

Comments and Suggestions for Authors

The authors describe five cases of non-infectious posterior uveitis studied by OCT-angiography (OCTA). They conclude that this method is useful for the diagnosis of ocular vascular lesions in this disease.

Comments:

  • A title such as "OCT-angiography in noninfectious uveitis: description of five cases and clinical applications" in my opinion would be clearer and more informative

we have made this change to the manuscript.

2)   The authors should specify in the background section of the abstract the unmet needs related to the study of uveitis that may justify adding the use of OCTA. They should also specify in this section that this is a series of 5 clinical cases with non-infectious posterior uveitis.

 We agree this is a great addition to the abstract: We have added: “While fluorescein angiography is still the gold standard for the diagnosis of posterior uveitis, it has limitations, and therefore it can be replaced by OCTA in some cases.”

And in Methods, there was the sentence: “ This case series describes 5 patients with posterior noninfectious uveitis and their description by OCTA.”

  • In the study, patients were studied with three different instruments for OCTA. It would be helpful to know if there are differences in the sensitivity and specificity of the results obtained with different equipment.

We have added in the Discussion in the section about limitations of the study, the differences in the sensitivity/ specificity of the results obtained.

“Several published studies have compared the 2 OCTA devices that we used, Heidelberg Spectralis OCT2 module and Zeiss Plex Elite 9000 Swept-Source OCT. Lu et al, have compared a smaller section 3x3mm and they showed that the foveal avascular zone area showed no significant difference across devices with an ICC of > 95%. However, there were statistically significant differences for SCP and DCP vessel densities. They found also a statistically significant difference for fractal dimension at the DCP layer (ref). Another study has found a sensitivity of 96.3% for the PlexElite device, 87.9% for the Heidelberg OCT2, for the detection of macular neovascularization (MN) in neovascular age macular degeneration (ref). However, in uveitis most of the MNVs are of type 2 and the same study found that 100% of this type of MNVs were detected by all OCTA devices, independently of the SD- or SS-technology. They found that undetectable/undetected MNV can represent up to 45% of the examination (ref).”

  • Patient 3 is described as having histoplasmosis uveitis. Since the authors consider patients with noninfectious uveitis, this point should be clarified

The case number 3 refers to a patient who had pseudo-histoplasmosis that corresponds to choroidal lesions that look like histoplasmosis choroiditis but the serology for histoplasmosis was negative. Usually, ocular pseudo-histoplasmosis responds well to systemic steroids and immunosuppressive treatment without any anti-infectious treatment. While it cause remains unknown, ocular pseudo-histoplasmosis responds to treatment like a non infectious disease.

  • Although the clinical cases are described in detail throughout the text, Table 1 should be completed with demographic data of the patients (age, sex etc.), bilaterality or monolaterality of uveitis, BCVA, current therapy, to make the reading of the table more informative.

It has been done.

  • Two patients were on hydroxychloroquine therapy. Given the retinal toxicity of this drug, a comment on this is useful

Case number 1. “We have added that the dose was 4.7mg/kg/d oral hydroxychloroquine since the last 5 years.” And that we have performed a multifocal ERG to rule out plaquenil toxicity : “A multifocal electrophysiology showed that the responses in the fellow eye, left eye were within normal limits indicating normal macula function. There is no generalized or focal decrease in response amplitude or implicit time suggestive of hydrochloroquine toxicity. In the right eye, the responses were reduced temporally consistent with the lupus retinopathy and consequent visual field deficit observed.”

Case number 2. We have added that the hydroxychloroquine was started 18 months ago that was too early to show retinal toxicity. We have also added : “SD-OCT was performed that showed unremarkable outer retinal layers, ruling out hydroxychloroquine toxicity; the 10-2 Humphrey visual fields was unreliable.”

7)   Authors should provide some data on patient monitoring (frequency of visits, any changes over time possibly detected by OCTA

We have articulated the important time points of follow up and significant pathology detected by OCTA at these instances in the case descriptions.

Case number 1, we added: “OCTA acquisition was continued at each visit every 3 to 4 months during 29 months to monitor the area of ischemic maculopathy and to guide treatment. No change in flow deficit was detected”.

Case number 2, ICG was performed at baseline and then OCTA was performed at 6-week follow-up to monitor the lesions. We have not performed any follow-up for that patient so far.

Case number 3, we stated that “The lesion is smaller on OCTA at 3-month follow-up.” We have added :” OCTA testing were repeated 2 monthly for a duration of 8 months afterwards and the images remained unchanged.”

Case number 4, we showed OCTA at 3 and 4 months follow-up (Figure 4).

Case number 5, we have no follow-up OCTA available in the chart.

8)      The advantage of OCTA over other methods that explore the retinal vasculature (fluorescein angiography, indocyanine angiography, enhanced-depth imaging-OCT) should be better commented. The authors should also comment on whether OCTA is sufficient alone or should necessarily be combined with other diagnostic methodologies in the cases studied.

In the discussion, we have stated, “OCTA plays an increasingly important role in the diagnosis and monitoring of treatment response in uveitis and can be used in combination with FA, ICG, and OCT. We recommend a multimodal approach to visualize the choroidal and retinal vasculature in inflammatory processes.”

9)      It is also useful to comment on whether this method might be useful also in infectious forms of uveitis

We agree, we have modified the Discussion for “Despite the limitations, we showed that OCTA plays an increasingly important role in the diagnosis and monitoring of treatment response in non infectious posterior uveitis….” and for :

“Of note, previous studies have showed that OCTA has the unique ability to demonstrate pathological flow impairment at the level of choriocapillaris in infectious posterior uveitis like in active tubercular SLC. Particularly, Agarwal et al showed that during longitudinal follow-up, in the multifocal type of SLC, the mean lesion area de-creased in all eyes compared with baseline”.

10) Approval of the study by the ethics committee should also be specified in the materials and methods section

“This study was conducted in accordance with the tenets of the Declaration of Helsinki and approved by University of Pittsburgh Medical Center’s (UPMC) Institutional Review Board (Number: Number: STUDY19030187 - Multimodal Imaging in Ophthalmology).” Is stated in the acknowledgements section, but we have added it in the materials section.

Reviewer 2 Report

The manuscript of Samyuktha Melachuri et al. describes interesting issues, since OCT / OCTA is increasingly used by ophthalmologists and has become one of the primary diagnostic tools. The selection of literature is correct. The logical structure and division of the manuscript are also appropriate. The language is comprehensible and correct.

However, the submitted manuscript is written in a very cursory and general way. The section on materials and methods is residual, and the sections on results and their discussion are cursory.

 Major comments:

1) According to the requirements of the Diagnostics Journal, “the article should report scientifically sound experiments and provide a substantial amount of new information.” Therefore, I wonder whether this manuscript can be considered a full text article, since only six patients were studied, one for each case.

2) The manuscript describes interesting issues, but is written in a very cursory and general way. The section on materials and methods is residual, and the sections on results and their discussion are cursory. Can the authors explain the novelty of their study with reference to publication doi:10.1136/bjophthalmol-2020-31688.

3) The main problem is the limited number of patients examined,1 for each case. This makes the results obtained more qualitative than quantitative. Thus, I have doubts about the statistical significance of the results obtained and their reproducibility. The results may vary between different patients, even with the same disease, so what guidelines can be developed based on the analysis of a single case? I am concerned that the conclusions presented are not adequately supported by the results obtained. The authors only described the images, but they can provide much more quantitative information (for example, see https://doi.org/10.1167/tvst.9.3.10, https://doi.org/10.1167/tvst.10.1.29)

4) Lines 75-81: What criteria were used for segmentation? What was their reason? Please provide a more detailed explanation and description in text.

5) Section 2 is written too vaguely and too general. The authors are asked to describe in detail the methodology adopted if these results are to be of any use to clinicians. In this section, the authors should add the schema of the proposed methodological approaches used in analysis of the OCTA data. This section should be further developed as it mainly allows tests to be repeated by other researchers and the measurement protocols developed to be verified.

6) Line 73: The authors used 4 different OCT systems? It should be more directly indicated? If so, please provide a comparison of the main parameters of the OCT systems used.

7) If 4 different OCTA systems are used,

8) What was the spectral range and central wavelength of the radiation source in the system used?

9) What was the A-scan/B-scan rate during the OCT examination? The speed with which a B-scan is collected depends on the A-scan or the line rate, but a higher A-scan rate results in lower sensitivity.

10) What was the FOV of the OCT system used?

11) What was the imaging depth of the OCT system used?

12) How many OCT images were registered for each patient?

13) line 82: 'en face structure' - which one? It should be clarified and specified.

14) line 83: „shadow” -– which one? It should be clarified and specified.

15) Line 86: 'OCTA signal attenuation', what level of attenuation (any range) was taken as corresponding to 'flow deficit'. Authors did not try to quantitatively characterise ROI by any image processing algorithms?

16) Did the authors analyse the speckles on the registered images?

17) Line 121: “deep capillary plexus (DCP) (not shown here)” – why? Why are not all the results analysed in the manuscript presented?

18) Lines 122-123: “flow deficit observed on OCTA was of similar size to the retinal ischemia detected on FA 2.5 months prior” – any quantity?

19) Any explanation in the caption of the figure of the regions of interest (circles, stars, arrows ect.) marked in the recorded images ?

20) No scale bars on the images presented. Please correct this?

21) I wonder if it would be more beneficial to a larger readership not to compare the OCTA images of each analysed patient with a healthy person (healthy control)

22) What is the difference in intensity between images D and E in figure 5 due to?

Minor comments: Please read carefully and check for formatting errors.

1) Section 2- 3.1: correct the text formatting.

2) Line 33: I suppose that the dot should be after the citation. Please correct all manuscripts.

3) Line 86: ““flow deficit.” - quotation marks after the dot.

4) Fig.4/Fig.5: Do the entire figure captions should be written in bold?

Author Response

OCTA Publisher Comments

Reviewer Panel

Diagnostics journal

March 5th, 2023

Dear Reviewers,

Thank you for your helpful comments and suggestions for our manuscript, Manuscript ID: diagnostics-2204641. Please find attached a revised version of the manuscript with Track Changes enabled. Your comments are also addressed as follows:

Comments and Suggestions for Authors

The manuscript of Samyuktha Melachuri et al. describes interesting issues, since OCT / OCTA is increasingly used by ophthalmologists and has become one of the primary diagnostic tools. The selection of literature is correct. The logical structure and division of the manuscript are also appropriate. The language is comprehensible and correct.

However, the submitted manuscript is written in a very cursory and general way. The section on materials and methods is residual, and the sections on results and their discussion are cursory.

 Major comments:

  • According to the requirements ofthe Diagnostics Journal, “the article should report scientifically sound experiments and provide a substantial amount of new information.” Therefore, I wonder whether this manuscript can be considered a full text article, since only six patients were studied, one for each case.

We appreciate your comments. Due to the large variety of posterior uveitis condition, we chose 5 representative cases of noninfectious posterior uveitis, each of them involving a specific anatomic layer: superficial capillary plexus, choriocapillaris, choroid. We have added a sentence to the first paragraph of Methods to explain this basis for case selection, and to overcome any concern that cases might have been selected arbitrarily.

We believe we have provided a substantial amount of information though we utilized only a few patient examples.

  • The manuscript describes interesting issues, but is written in a very cursory and general way. The section on materials and methods is residual, and the sections on results and their discussion are cursory. Can the authors explain the novelty of their study with reference to publication DOI:10.1136/bjophthalmol-2020-31688.

In uveitis, we provided the new information that en face structure gave the same information as OCTA in the detection and monitoring of inflammatory choroidal neovascular membranes under appropriate treatment. Similarly, en face structure also gave the same level of information than OCTA in choriocapillaris flow reduction or ischemia defects in APMPPE lesions.

We used the terminology reported in the publication : “Pichi F, Salas EC, D de Smet M, Gupta V, Zierhut M, Munk MR. Standardisation of optical coherence tomography angiography nomenclature in uveitis: first survey results. Br J Ophthalmol. 2021 Jul;105(7):941-947. doi: 10.1136/bjophthalmol-2020-316881. Epub 2020 Jul 29. PMID: 32727731.”

We were not meant to introduce new terminologies in the current paper.

The paper by Pichi et al. focuses on standardizing OCTA nomenclature. Our manuscript is focused more on worked case examples in which the terminology is applied. Naturally, there is some overlap, but there are sufficient differences in emphasis to constitute new information.

An important concept explained in our paper, which is absolutely essential to the correct interpretation of reduced flow signal on OCTA, is that distinguishing between masking and genuinely reduced flow requires side by side comparison with the equivalent slice from the structural OCT.

We have added the following text to the Methods:

“We applied the Pichi et al. nomenclature:1/ the terms ’flow deficit’ and ’non-detectable flow signal’ to describe abnormal OCTA flow signal secondary to slow flow and to vessels displacement respectively; 2/ the terms ’loose’ and ’dense’ to describe the appearance of inflammatory choroidal neovascularization, and 3/ the percentage of flow signal decrease to measure OCTA ischemia.”

We have added the following text to Discussion.

We found that the Pichi et al. nomenclature was effective for describing the findings: the terms ’flow deficit’ to describe abnormal OCTA flow signal secondary to slow flow to measure retinal ischemia in the case of occlusive retinal vasculitis (case number 1), and ’non-detectable flow signal’ to describe vessels displacement secondary to choroidal granulomas, multifocal choroiditis or placoid lesion in APMPPE (cases number 2, 4 & 5). We used the terms’dense’ to describe the appearance of inflammatory choroidal neovascularization (case number 3).

  • The main problem is the limited number of patients examined,1 for each case. This makes the results obtained more qualitative than quantitative. Thus, I have doubts about the statistical significance of the results obtained and their reproducibility. The resultsmay vary between different patients, even with the same disease, so what guidelines can be developed based on the analysis of a single case? I am concerned that the conclusions presented are not adequately supported by the results obtained. The authors only described the images, but they can provide much more quantitative information (for example, see https://doi.org/10.1167/tvst.9.3.10https://doi.org/10.1167/tvst.10.1.29)

Danuta M. Sampson, Noha Ali, Alex Au Yong, Rumaanah Jeewa, Saumya Rajgopal, Deepaysh D. C. S. Dutt, Sharaf Mohamed, Shehata Mohamed, Alex Hansen, Moreno Menghini, Fred K. Chen; RTVue XR AngioVue Optical Coherence Tomography Angiography Software Upgrade Impacts on Retinal Thickness and Vessel Density Measurements. Trans. Vis. Sci. Tech. 2020;9(3):10. doi: https://doi.org/10.1167/tvst.9.3.10.

Paula K. Yu, Andrew Mehnert, Arman Athwal, Marinko V. Sarunic, Dao-Yi Yu; Use of the Retinal Vascular Histology to Validate an Optical Coherence Tomography Angiography Technique. Trans. Vis. Sci. Tech. 2021;10(1):29. doi: https://doi.org/10.1167/tvst.10.1.29.

  •  

We have added choroidal thickness data throughout the manuscript. We have added in Methods: “The choroidal thickness (CT) measurements were obtained from manual segmentation of OCT B-scans at the fovea for cases number 1,2,4 and 5 and at the location of the choroiditis lesion (for case 3). Follow-ups CT measurements were available for cases number 1, 3 and 4.” Page 3 line 127.

In Table 1, we have added the choroidal thickness measurements for each case.

In Results, we have added “The mean CT was not significantly reduced at last follow-up compared with baseline (mean 292um ±132 at baseline vs mean 260um ± 108) (P = 0.75).”  Page 4 line 143.

We have added a supplemental figure 8 as an example for CT measurements. Legend: Case 1 Ischemic retina in lupus retinopathy. A. The choroidal thickness measurements were obtained from manual segmentation of OCT B-scans at the fovea at baseline, +11 months, +21 months and +32 months.

A substantial volume of literature now exists on OCTA “vascular density” but this has not resulted in any useful clinical paradigms in which patient care is reproducibly guided by “vascular density” measurements.

Paula Yu’s paper explains that OCT/OCTA cannot resolve smaller caliber vessels and supports the view we hold that “vascular density” does not actually represent vascular density. "VD” calculations amount to calculating ratios of bright and dark pixels, but they can tell us nothing about vascular diameter, vascular flow, vascular engorgement or, indeed vascular density, because small vessels are too small for OCT to resolve.

Danuta Sampson’s paper explains that VD calculations can be influenced by software upgrades and thus raises concerns about reproducibility.

We do not use OCTA “VD” calculations clinically.

  • Lines 75-81: Whatcriteria were used for segmentation? What was their reason? Please provide a more detailed explanation and description in text.

We relied on automated segmentation proprietary to each device (closed source). These are based on pre-set boundary curves, such as vitreo-retinal interface, outer retina, RPE, choriocapillaris. If there were segmentation errors in a particular curve set, we chose the next closest curve set which did not have segmentation errors and which approximated the contours of the tissue feature of interest. We were then able to adjust the global z-position of this curve set to window in on the tissue feature of interest.

We have added an explanation to Methods.

  • Section 2 is written too vaguely and too The authors are asked to describe in detail the methodology adopted if these results are to be of any use to clinicians. In this section, the authors should add the schema of the proposed methodological approaches used in analysis of the OCTA data. This section should be further developed as it mainly allows tests to be repeated by other researchers and the measurement protocols developed to be verified.

The presence or absence of OCTA features of clinical interest was based on the paper by Pichi and colleagues.

  • Line 73: The authorsused 4 different OCT systems? It should be more directly indicated? If so, please provide a comparison of the main parameters of the OCT systems used.

We used 2 systems: Zeiss Plex Elite (swept source) using 1060 nm, and

Heidelberg OCT2 using 840 nm. Described in Methods, page 2 line 83 in manuscript.

  • If 4 different OCTA systemsare used, What was the spectral range and central wavelength of the radiation source in the system used?

Two systems were used.

Zeiss Plex Elite (swept source) using 1060 nm, and

Heidelberg OCT2 using 840 nm.

We do not know the spectral range of either of the light sources.  

This information was added page 2 line 83 in manuscript.

Spectral domain OCTA (SD-OCTA) utilizes a near-infrared super luminescent diode with a wavelength of ~ 840 nm as a light source, and a spectrometer as a detector. Swept Source OCTA (SS-OCTA) utilizes a tunable swept laser with a wavelength of 1060 nm as a light source and a single photodiode as a detector.

9) What was the A-scan/B-scan rate during the OCT examination? The speed with which a B-scan is collected depends on the A-scan or the line rate, but a higher A-scan rate results in lower sensitivity.

The Zeiss Plex Elite operates at 100 kHz; our device has not yet had the 200,000 kHz upgrade.

The Heidelberg operates at 85 kHz.

10) What was the FOV of the OCT system used?

For each image, we have now specified the field of view in the caption.

11) What was the imaging depth of the OCT system used?

We have not found that information for the 2 OCT systems used.

12) How many OCT images were registered for each patient?  We did not understand this question.

13) line 82: 'en face structure' - which one? It should be clarified and specified. We added :” we used the corresponding en face structural OCT image to evaluate whether a shadow could be detected”.

14) line 83: „shadow” -– which one? It should be clarified and specified. Shadowing is caused by an obstacle of light penetration through the retina which is obscured by an overlying pathology. This reduction in signal is termed shadowing and can be misinterpreted as reduced flow.

Page 3 line 106 in manuscript.

15) Line 86: 'OCTA signal attenuation', what level of attenuation (any range) was taken as corresponding to 'flow deficit'. Authors did not try to quantitatively characterize ROI by any image processing algorithms?

The focus of this manuscript is on the utility of commercially available OCTA in managing uveitis patients. Quantification of signal attenuation using ROI is not a commercially available workflow. Flow deficit was therefore determined subjectively by inspecting the images.

16) Did the authors analyze the speckles on the registered images? No

17) Line 121: “deep capillary plexus (DCP) (not shown here)” – why? Why are not all the results analyzed in the manuscript presented?

We have added one Supplemental Figure 6 with “deep capillary plexus on OCTA and en face”

18) Lines 122-123: “flow deficit observed on OCTA was of similar size to the retinal ischemia detected on FA 2.5 months prior” – any quantity? Flow deficit was therefore determined subjectively by inspecting the images.

19) Any explanation in the caption of the figure of the regions of interest (circles, stars, arrows ect.) marked in the recorded images ?

Thank you, We added the regions of interest in legend for each figure.

20) No scale bars on the images presented. Please correct this?  For each image, we have now specified the field of view in the caption.

21) I wonder if it would be more beneficial to a larger readership not to compare the OCTA images of each analyzed patient with a healthy person (healthy control)

Thank you for the suggestion. We have added a supplemental image that include all different layers: superficial, deep capillary plexuses, choriocapillaris, choroid to compare with each analyzed patient (Supplemental Figure 7).

Legend: Healthy control. Zeiss PLEX ® Elite 9000 OCTA (12x12 mm) of superficial capillary plexus (A), deep capillary plexus (D), choriocapillaris (G), choroid (J) and the corresponding inner en face structure (B, E, H, K) and corresponding  Optical Coherence Tomography (OCT) B-scans (C, F, I, L).

22) What is the difference in intensity between images D and E in figure 5 due to?

For some reason the OCT image done by Spectralis Heidelberg SD-OCT (image D) had a better intensity than the OCT done by the Zeiss PLEX ® Elite 9000 at follow-up (image E). Different machines. We added in legends the names of different devices.

Minor comments: Please read carefully and check for formatting errors.

  • Section 2- 3.1: correct the text formatting.

We have corrected the formatting.

2) Line 33: I suppose that the dot should be after the citation. Please correct all manuscripts.

Thank you for your insight. We have corrected this.

  • Line 86: ““flow deficit.” - quotation marks after the 

Thank you, we have made this change.

  • 4/Fig.5: Dothe entire figure captions should be written in bold?

Thank you, we have made this change.

Round 2

Reviewer 1 Report

I thank the authors to have fully answered to my questions 

Author Response

Thank you  for your meticulous and thoughtful review, your helpful comments and suggestions for our manuscript.

Reviewer 2 Report

I would like to thank the authors for the clarifications they sent and the corrections they made to the manuscript. However, before acceptance for publication, some additional corrections are necessary:

Major comments:

1) According to their statement: “Swept Source OCTA (SS-OCTA) utilizes a tunable swept laser with a wavelength of 1060 nm as a light source and a single photodiode as a detector.” – Based on the online Zeiss technical specifications of this OCT system : https://www.zeiss.com/content/dam/Meditec/downloads/pdf/ari-network-download/plex-elite-brochure_en_31_020_0001v_us_31_020_0001v.pdf

I have discovered that the light source used is a tunable laser with a centre wavelength between 1040 and 1060 nm. Please correct this information.

2) According to my previous question What is the difference in intensity between images D and E in figure 5 due to?

Authors answer: For some reason the OCT image done by Spectralis Heidelberg SD-OCT (image D) had a better intensity than the OCT done by the Zeiss PLEX ® Elite 9000 at follow-up (image E). Different machines. We added in legends the names of different devices.

Does not the authors think it might have something to do with the absorption spectrum of water and hemoglobin (oxy- and deoxyhaemoglobin)? Sensitivity of the detector used?

3) According to the authors’ statement: “The Zeiss Plex Elite operates at 100 kHz; our device has not yet had the 200,000 kHz upgrade. The Heidelberg operates at 85 kHz”. Please indicate directly that this is the frequency of A scans.

Minor: Correct typographical errors in the micrometre notation, for example line 160: not “um” but "µm” etc.

Author Response

Major comments:

1) According to their statement: “Swept Source OCTA (SS-OCTA) utilizes a tunable swept laser with a wavelength of 1060 nm as a light source and a single photodiode as a detector.” – Based on the online Zeiss technical specifications of this OCT system : https://www.zeiss.com/content/dam/Meditec/downloads/pdf/ari-network-download/plex-elite-brochure_en_31_020_0001v_us_31_020_0001v.pdf

I have discovered that the light source used is a tunable laser with a centre wavelength between 1040 and 1060 nm. Please correct this information.

Author: thank you we have modified line 81: "using  a swept-source tunable laser: with a center wavelength between 1040 nm and 1060 nm as the optical source"

2) According to my previous question What is the difference in intensity between images D and E in figure 5 due to?  

Authors answer: For some reason the OCT image done by Spectralis Heidelberg SD-OCT (image C) had a better intensity than the OCT done by the Zeiss PLEX ® Elite 9000 at follow-up (image E). Different machines. We added in legends the names of different devices.

Our Answer:

Although swept source-optical coherence tomography (SS-OCT) technology offers improvements in visualizing the vitreous, retina, choroid, and sclera as compared as the Spectral Domain (SD)-OCT technology by increased scan speeds, decreased signal attenuation, and deeper tissue penetration in optimal patients, the technology can still be confounded by poor signal strength. Common causes include media opacity, ocular surface disease, reduced tear film quality, miotic pupils and image or motion artifacts. This might be the cause in this case. We excluded the hypotheses of media opacity and ocular surface disease because the patient had a clear lens and a clear cornea.

Therefore, we modified as follows:

"Due to poor signal strength (either miotic pupil or motion artifact), the OCT image done by Spectralis Heidelberg SD-OCT (image D) had a better intensity than the OCT done by the SS-OCT Zeiss PLEX ® Elite 9000 at follow-up (image F)." 

Does not the authors think it might have something to do with the absorption spectrum of water and hemoglobin (oxy- and deoxyhaemoglobin)? Sensitivity of the detector used?

Authors answer: We think that the difference in intensity is caused by poor signal strength (either miotic pupil or motion artifact) because usually the intensity of SSO-OCT is better quality.

SS-OCT has improved acquisition speed, volume and depth of ocular tissue measurements compared with SD-OCT technology (ref: Drexler W, Liu M, Kumar A, et al. Optical coherence tomography today: speed, contrast, and multimodality. Journal of biomedical optics. 2014;19(7):071412). This technology has the potential to provide excellent image resolution from the posterior hyaloid face through the choroid without the need for multiple image averaging or loss of internal retinal layer image quality when viewing deeper retinal or choroidal structures, as is the case of enhanced depth imaging functions with SD-OCT.

SD-Spectralis vs SS-Plex Elite 9000:

Scan speed (A-scan/sec): 8500 vs 100,000

wave lenghth (µm): 870 vs 1,000

scan density: 512 vs 500

lateral resolution (µm): 5.7 vs 10-20

axial resolution (µm): 3.9 vs 1.95 (digital) / 6.3 (optic function)

3) According to the authors’ statement: “The Zeiss Plex Elite operates at 100 kHz; our device has not yet had the 200,000 kHz upgrade. The Heidelberg operates at 85 kHz”. Please indicate directly that this is the frequency of A scans.

Thank you, we modified: "The Zeiss Plex Elite operates at a scan speed of 100, 000 scans/sec, our device has not yet had the 200,000 Hz upgrade. The Heidelberg Spectralis ® OCT2 operates with a scan rate of 85, 000Hz."

Minor: Correct typographical errors in the micrometre notation, for example line 160: not “um” but "µm” etc.

Done, thank you.